# A Study on Tensile Behavior According to the Design Method for the CFRP/GFRP Grid for Reinforced Concrete

**DOI:** 10.3390/ma15010357

**Published:** 2022-01-04

**Authors:** Jin Sung Kim, Seong Jong Kim, Kyoung Jae Min, Jung Chul Choi, Hwa Seong Eun, Bhum Keun Song

**Affiliations:** Convergence Research Division, Korea Carbon Industry Promotion Agency, 110-11 Banryong-ro, Jeonju 54853, Korea; kjs3428@kaist.ac.kr (J.S.K.); sjongkim@kcarbon.or.kr (S.J.K.); bluemy@kcarbon.or.kr (K.J.M.); happycjc@kcarbon.or.kr (J.C.C.); hs_eun@kcarbon.or.kr (H.S.E.)

**Keywords:** CFRP grid, GFRP gird, reinforced concrete, VRHP mortar, tensile behavior, design method, finite element analysis

## Abstract

In the present study, fiber-reinforced plastics (FRP) grid-reinforced concrete with very rapid hardening polymer (VRHP) mortar composites were fabricated using three types of design methods for the FRP grid (hand lay-up method, resin infusion method, and prepreg oven vacuum bagging method), along with two types of fibers (carbon fiber and glass fiber) and two types of sheets (fabric and prepreg). The FRP grid was prepared by cutting the FRP laminates into a 10 mm thick, 50 mm × 50 mm grid. The tensile behavior of the FRP grid embedded in composites was systematically analyzed in terms of the load extension, fracture mode, partial tensile strain, and load-bearing rate. The CFRP grid manufactured by the prepreg OVB method showed the best tensile behavior compared to the CFRP grid manufactured by the hand lay-up and resin infusion methods. The load-bearing of each grid point was proportional to the height from the load-bearing part when reaching the maximum tensile load. In addition, finite element analysis was conducted to compare the experimental and analysis results.

## 1. Introduction

Reinforced concrete is the most widely used construction material for structures [1]. The steel rebars embedded in the concrete to reinforce the tensile load of concrete are characterized by thermal expansion behavior almost identical to that of concrete [2]. In reinforced concrete structures, microcracks in the structure due to physical impact, aging, fire, corrosion, etc. are the main causes of internal corrosion of steel rebars [3]. Corroded steel rebars cause structural durability and safety problems [4]. Methods such as reducing the penetration of corrosive ions in concrete [4] and covering the steel rebars [5] have been studied to prevent corrosion of the steel rebar. Among them, the fiber-reinforced plastics (FRP) grid has received attention as a material that can replace steel rebars [6]. The FRP grid is a composite material with high corrosion resistance and chemical resistance compared to steel rebar [7].

The FRP grid is manufactured by stacking FRP composites into a grid shape. FRP composite is a lightweight, high-strength composite material made by mixing reinforced fibers that supports load, and it can be made into a matrix that absorbs shock and fixes reinforcement [8]. In addition, various strength designs are possible, and it has the advantages of less thermal deformation and no rust [9]. However, there are still disadvantages, such as high raw material price, high temperature vulnerability, and lack of automation of the manufacturing method [10]. FRP composites include glass fiber reinforced plastics (GFRP), basalt fiber reinforced plastics (BFRP), aramid fiber reinforced plastics (AFRP) and carbon fiber reinforced plastics (CFRP). Glass fiber made from thin glass has good cost effectiveness, good impact resistance, and good fire resistance, but its weight reduction efficiency is low [11]. Basalt fiber made by melting basalt at high temperature is an eco-friendly fiber that does not generate carcinogens, and has good mechanical properties compared to glass fiber, but its weight reduction efficiency is low. Aramid fiber has a flame retardant, good elongation, and impact resistance, but is expensive and has a poor resin adhesion rate [12]. Carbon fiber made by burning organic fiber at high temperature has high strength and rigidity and high weight reduction efficiency, but it is weak against impact and expensive [13]. In addition, due to its low thermal deformation, high heat resistance, chemical resistance, and electrical conductivity, carbon fiber is used for various applications such as aerospace, automobiles, structural materials, and biomaterials [14].

The manufacturing method of FRP is selected considering the purpose and cost of FRP composites. In FRP molding, the final product is manufactured by directly molding the fabric or through secondary molding after fabricating an intermediate material such as prepreg. Representative direct molding methods include hand lay-up and resin infusion methods. The hand lay-up method is used for simple and low-volume products, and is applied for boat shells, swimming pools, and bathtubs. The resin infusion method is mainly used to make large structures such as wind blades. In the prepreg manufacturing method, a final product is manufactured by compaction of pre-prepared prepregs mainly through the oven vacuum bagging method [15,16,17,18,19,20]. The detailed manufacturing method of the above three methods will be discussed in Section 3.

The manufactured FRP grid is used to supplement the tensile performance of concrete as a substitute for rebar in reinforced concrete. In recent research into CFRP grid reinforced concrete, various methods have been proposed to improve the properties of the CFRP grid and concrete. For instance, studies have been conducted to use engineered cementitious composite (ECC) as a part of concrete in the tensile region of concrete. ECC is a cement matrix incorporating PVA fibers within 2% volume fraction, with an ultimate tensile strain of 3~7% [21]. Zhang et al. [22] synthesized CFRP-grid-reinforced ECC with improved marine environment durability. The results showed that the proposed CFRP-grid-reinforced ECC composites improve the maximum compressive load and cracking load compared to the existing beam. In that study, a new hybrid CFRP-reinforced concrete beam combining CFRP grids and ECC layers was proposed, and the flexural performance was investigated. Zhu et al. [23] experimentally examined the mechanical behavior of CFRP-ECC on short concrete columns. The results indicated that the main cause of column failure was the destruction of the embedded CFRP grid. In that study, a mechanism for the stress–strain relationship of a concrete column composed of CFRP-ECC was presented, and its validity was verified through analysis results.

Studies have been investigated to optimize the CFRP grid shape for reinforced concrete. Guo et al. [24] fabricated CFRP-reinforced concrete with polymer cement mortar (PCM). The bonding behavior and stress transfer mechanism of four types of grids with different intervals and lengths were discussed. In that study, the effect of the number of grid points and the relationship between the horizontal and vertical grids and the bonding behavior of the CFRP grid were investigated. Wang et al. [25] examined the pull-out test of a single grid point and CFRP grid embedded in the concrete. The results showed that the tensile strength of the CFRP grid embedded in the concrete was less than half that of the single grid. In that study, the fracture mechanism of composite specimens was investigated based on the bonding behavior between concrete, the CFRP grid, and the resistance behavior of mortar.

In the present study, three types of CFRP grids using three design methods (hand lay-up method, resin infusion method, and prepreg oven vacuum bagging (OVB) method) and GFRP using the prepreg OVB method were fabricated. The fabricated FRP grids were incorporated between concrete and very rapid hardening polymer (VRHP) mortar. The main essences of the present study are as follows: (1) present a design guideline of CFRP and GFRP grids, (2) investigation of tensile behavior of FRP grid incorporated in FRP grid/concrete/VRHP mortar composites, (3) a study on tensile strain distribution of FRP grids and (4) comparative analysis with experimental results through finite element analysis. The tensile behavior of the FRP grid embedded in the concrete/VRHP mortar composites was analyzed. The experimental results of load extension, fracture mode, partial tensile strain, and load-bearing rate were systematically investigated to evaluate the FRP grid embedded in composites. In addition, finite element analysis was conducted to compare the experimental and analysis results.

## 2. Experimental Program

In this study, the tensile behavior of the FRP-grid-reinforced concrete composites was investigated. FRP laminates were fabricated using three manufacturing methods. The hand lay-up method, resin infusion method, and prepreg OVB method were employed using carbon fiber fabric, carbon fiber prepreg, and glass fiber prepreg, respectively. Four types of FRP grid were manufactured by cutting the FRP laminates into 50 mm × 50 mm grid intervals with a width of 10 mm. Table 1 and Table 2 notice the FRP laminate properties and type of FRP grid specimens. Specimen codes are defined as follows: The first letter ‘H’ stands for hand lay-up method, ‘I’ stands for resin infusion method, and ‘P’ stands for prepreg OVB method. The second letter ‘C’ stands for carbon fabric, and ‘G’ stands for glass fabric. The tensile behavior of the FRP grid was measured at five grid points of the FRP grid, and the pull-out test was conducted on three samples of each type.

The FRP-grid-reinforced concrete composites were designed based on research conducted by Wang et al. [25]. Figure 1 indicates the design of the FRP grid/concrete/VRHP mortar composites. The mix proportions of the concrete specimens are given in Table 3. River sand (0.06~2 mm) and gravel (9.5~37.5 mm) were used as fine and coarse aggregates, respectively. Concrete specimens were cured for 28 days at room temperature conditions. After curing of the concrete specimen was completed, the surface was smoothed using sandpaper. The FRP grid was attached on a concrete specimen by a surface attachment method without using a rivet anchor because the overall size of the composite was not large. [24]. Strain gauges were installed at P1~P5 grid point of FRP grid. The strain gauge was coated with silicone bond and rubber tape to prevent external impact. VRHP mortar was overlayed on the composites after attaching the FRP grid to the fabricated concrete specimen. VRHP mortar is one of the concrete repair and reinforcement materials characterized by rapid initial strength development, high fluidity, and high concrete adhesion [26]. VRHP mortar filled the protruding part of the FRP grid due to its high fluidity and fixed the FRP grid to the concrete specimen. VRHP mortar was prepared by mixing water in an 18% weight ratio of mortar. VRHP mortar was cured for 3 days at room temperature conditions. The compressive strengths of concrete specimen and VRHP mortar were 40.26 and 43.87 MPa, respectively. The compressive strength of each specimen was measured by KS F 2405 test under 20 ± 15 °C, 65 ± 20% R.H conditions. The reliability of this reinforcing method can be secured through the investigation of the tensile behavior and stress transmission mechanism of the composites by combining the concrete–FRP grid–VRHP mortar. Figure 2 shows the fabrication process of composites.

After curing the VRHP mortar, all samples of FRP grid/concrete/VRHP composites were tested using the pull-out test. The pull-out test process was designed based on research conducted by Wang et al. [25]. Figure 3a shows the schematic of a jig for holding the composites during the pull-out test. By connecting the exposed FRP grid on the jig with the INSTRON 5982 (Instron Co., Norwood, MA, USA), a tensile load was applied only to the FRP grid, while the composite was fixed by the jig. The pull-out test setup was installed as shown in Figure 3b. The tensile load was applied to the composites at a rate of 2 mm/min during the pull-out test. The strain gauge and universal testing systems (UTM) data were collected using a connected computer.

## 3. Design Guidelines for CFRP/GFRP Gird

### 3.1. Hand Lay-Up Method

The hand lay-up method is a method in which a worker manually laminates a textile fabric to form a product. It is a manufacturing method with advantages of low investment cost, relatively easy construction, simple required equipment, and fast curing time. However, the physical properties of the specimen are affected by the production capacity of the operator, and in order to achieve uniform quality, the operator’s skill is required, and the appearance of the specimen is not attractive due to the influence of manual work. In addition, there are problems such as the removal of air bubbles between the fabric and the resin and the optimization of the fiber content. The hand lay-up method is mainly used to produce a small number of products, not by mass production, and is used for boat shells, swimming pools, and bathtubs. In this study, the hand lay-up method was implemented as follows (Figure 4):(1)Prepare a 500 × 1000 mm^2^ bi-directional carbon plain weave fabric (0°/90°, 200 g/m^2^). Carbon fiber fabrics were purchased from Toray company (Tokyo, Japan).(2)Apply it to the mold surface using a release agent before lamination. In this experiment, a liquid release agent was used for uniform application.(3)Stir the epoxy resin. For the resin used in this experiment, mix KINETIX R118 infusion resin and H120 infusion hardener in a ratio of 100:25 and mix for 90 s.(4)After placing peel ply on the mold, impregnate it with resin. Peel ply facilitates the escape of the specimen and smooths the surface of the specimen. It also serves to prevent the inflow of external air bubbles.(5)On the peel ply, impregnate one piece of textile fabric. A brush or roller is used as an impregnation tool.(6)Finally, impregnate the peel ply and cover it with a release film to prevent the inflow of external air bubbles prior to addition; finish the preparation of the specimen.(7)After curing for 24 h, remove the peel ply to degrease the laminated specimen, then cut it according to the desired grid size to manufacture the FRP grid.(8)Hand lay-up specimens require the skill of the operator to remove air bubbles and uniformly impregnate the resin when impregnating the textile fabric.

### 3.2. Resin Infusion Method

The resin infusion process is one of the methods developed to replace the autoclave molding process mainly used for molding high-performance composite materials. The autoclave process requires expensive equipment that applies high temperature and high pressure; thus, the manufacturing cost is high, and it is not suitable for molding large structures. Conversely, the resin infusion process is a method that does not require expensive equipment, such as an autoclave. In addition, it is possible to mold products with low porosity and uniform quality compared to hand lay-up, a low-cost method. In addition, due to the short resin filling time, it is easy to manufacture large structures, and it is possible to mold products with complex shapes. However, it has the disadvantage of taking a long time to pre-process the product.

The resin infusion process is a molding method that manufactures products by laminating fibers or fabrics onto a mold, having a desired shape, generating a pressure difference with the internal air pressure through a vacuum, injecting resin into the fabric due to the pressure difference, and then curing it. The filling time of the resin is affected by the pressure, viscosity, and permeability and by the distance the resin moves. The amount of resin injected can be controlled using a pressure gradient, and the temperature at which the resin viscosity is minimized should be applied. Since the viscosity of the resin decreases as the temperature rises due to the reaction between the main material and the binder, the resin must be changed when it exceeds a certain temperature. In addition, it is possible to reduce the resin filling time by maintaining a high laminate permeability coefficient of the material and applying the optimal arrangement of the resin injection line according to the guideline setting of the mesh. In this study, the resin infusion method was implemented as follows (Figure 5):(1)Prepare a 500 × 1000 mm^2^ bi-directional carbon plain weave fabric (0°/90°, 200 g/m^2^). Carbon fiber fabrics were purchased from Toray company (Japan).(2)Apply to the mold surface using a release agent before lamination. In this experiment, a liquid release agent was used for uniform application.(3)Stir the epoxy resin. In this experiment, Resoltech resin 1050 and Resoltech hardener 1056 were mixed in a ratio of 100:35 and mixed for 90 s.(4)After spraying the spray adhesive onto the mold, apply a peel ply.(5)Lay each fiber fabric on the peel ply one by one using spray adhesive.(6)On the laminated textile fabric, laminate in the order of peel ply, release film, and mesh. The mesh facilitates resin fluidity in the longitudinal direction of the product such that the resin can be filled. The resin is cut 10–20% shorter in the longitudinal direction to form a break line.(7)Install omega flow and resin ports on both sides of the fabric in the longitudinal direction. Omega flow plays a role in smoothly reaching the desired length of the resin. The resin port is inserted by cutting the omega flow and is fixed with sealant to prevent leakage.(8)Lay the breather on the mesh, fix the bagging film around the fabric using sealant, and hold the vacuum twice. The breather plays the role of evenly distributing the vacuum pressure, preventing vacuum bag leakage, and holding the vacuum twice for uniformity of specimen thickness.(9)Inject the resin through the resin port and proceed until it finally arrives in the longitudinal direction of the product. Check the temperature of the resin during injection. As the temperature increases, the viscosity of the resin decreases; thus, when the temperature rises above a certain temperature, new resin is injected.(10)After resin injection is completed, after curing for 24 h, remove the peel ply to strip the laminated specimen, then cut it to the desired grid size to manufacture the FRP grid.

### 3.3. Prepreg OVB Method

In this method, a pre-impregnated material is used, unlike for the previous fabric. Prepreg is an intermediate material for manufacturing a composite material in the form of a sheet by pre-impregnating fibers or fabrics with resin. Compared to fabrics, it does not require resin impregnation; thus, it is possible to make products with high workability and clean appearance. Prepregs should be kept sealed at a low temperature, and it is recommended to minimize the exposure time to room temperature by exposing them directly before cutting. The oven vacuum bagging method is a type of autoclave method, and similar to resin infusion, it is a useful method for forming thin composite products within 10 plies with inexpensive equipment. In this study, the prepreg OVB method was performed as follows (Figure 6):(1)After attaching the peel ply through the air adhesive, laminate an adhesive film and one sheet of prepreg.(2)After attaching the first sheet of prepreg to the mold, laminate the release film and vacuum film.(3)After applying the breather, hold the vacuum injector and perform vacuum compaction at 50 degrees for 30 min.(4)After compaction, remove the release film, vacuum film, and breather, and then stack 2~3 ply of prepreg according to thickness and repeat steps 2 and 3. For complete adhesion between ply and removal of interstitial pores, do not laminate at once.(5)After lamination is complete, perform compaction at 50 °C for 30 min, hardening at 120 °C for 2 h, then strip and cut to the desired grid size to manufacture the FRP grid.

## 4. Results and Discussion

### 4.1. Load-Extension of CFRP/GFRP Grid

To evaluate the tensile behavior of the FRP grid embedded in FRP grid/concrete/VHPR mortar composites, the composite pull-out test was conducted. Figure 7 shows the load extension results of the FRP grid embedded in composites, and Table 4 shows the average values of the maximum tensile strain, maximum tensile stress, and maximum load of each specimen. All FRP grid specimens linearly increased displacement as the load increased; the load extension data measured after the maximum load are not included in the results. The experimental results indicate that the FRP grid was destroyed in the elasticity section, which shows the linear relationship between load and extension. The tensile behavior of the FRP grid in composites was lower than that of the FRP laminate used in the tensile strength test. The bending stress on the FRP grid occurring by the composite fabrication process and the pull-out test set-up affected the deformation of the FRP grid. In the composite fabrication process, the bending stress of the FRP grid may occur due to the separation caused by incomplete bonding between the concrete and FRP grid. In addition, the bending load may be applied to the FRP grid by a slight vertical slope of the composites due to unstable coupling between the designed jig and the composites. However, in actual concrete structures, depending on the shape of the structure and the direction of the load, a situation in which bending stress is applied to the FRP grid incorporated into the concrete often occurs. Therefore, the results of this experiment can be used meaningfully, as the FRP grid in the concrete is exposed to a complex situation where both tensile and bending loads are applied.

The maximum displacements for H.C, I.C, P.C and P.G were 14.09, 14.45, 15.49, and 8.24 mm, respectively. The maximum loads applied to each specimen were 13.23, 15.54, 20.02, and 7.43 kN, respectively. The results show that there is a difference in the tensile behavior of the FRP grid embedded in composites depending on the fabrication method, even if an identical carbon fiber is used. When comparing the H.C specimen and the I.C specimen, the maximum displacement of the I.C specimen increased by 6.96% and the maximum load by 17.43% compared to the H.C specimen. Even if the fabric type and the number of sheets laminated were identical, the tensile behavior of the CFRP grid varied depending on the fabrication process method and fiber impregnation rate. The fiber impregnation rate of the resin infusion method was 16% higher than that of the hand lay-up method. In the case of the resin infusion method, the resin was impregnated for a long time through the mesh by the vacuum bag process. Conversely, in the case of the hand lay-up method, the excess resin was not completely removed because the resin was impregnated by hand. In addition, since the vacuum bag process was not conducted, the fiber impregnation rate of the specimen manufactured through the hand lay-up method was relatively low. This is the reason the specimen manufactured by the hand lay-up method is thicker than the specimen manufactured by the resin infusion method, although the identical number of sheets was laminated. The resin and carbon fibers in the CFRP composites behaved together under the tensile load applied to the composites. Since the tensile strength of carbon fiber was relatively higher than that of resin, the higher the resin content, the more the tensile strength of CFRP is negatively affected.

The P.C specimen manufactured by the prepreg OVB method showed 15.68% and 8.14% higher displacements and 56.38% and 7.42% higher maximum loads compared to the H.C and I.C specimens, respectively. The difference in the material properties of the fabric and the prepreg affected the tensile behavior of the CFRP grid. The resin impregnation process is different between the resin infusion method and the prepreg OVB method. The resin infusion method is a method of impregnating resin in a slurry state thinly through a vacuum bag process and injecting it into the fabric. Although the resin infusion method is an excellent way to reduce excess resin, the adhesive film does not generate excess resin. In addition, since the prepreg sheet is also pre-impregnated with resin to fabricate a thin sheet, excess resin is not generated. The prepreg OVB method and the resin infusion method both press the fabric and the sheet through a vacuum bag process, but in the case of the resin infusion method, the entire composite undergoes the vacuum bag process twice, whereas the prepreg OVB method uses the vacuum bag process every 2~3 sheets. The sheets are closely pressed against each other. The H.C specimen showed a larger difference in tensile behavior than the I.C specimen. This is because the hand lay-up method generates relatively more excess resin and has a low adhesion between sheets.

The P.C specimen showed 88.74% higher tensile displacement and 179.66% higher maximum tensile load than the P.G specimen. Although an identical type of sheet and fabrication method were used, a definite difference in the tensile behavior of the grid occurred due to the difference in fiber types. The difference in fiber type had more significant effect on the difference in tensile behavior of the FRP grid than the difference in sheet shape and fabrication method. Therefore, in the FRP grid design process, the choice of fiber type should be considered more than the sheet type and process method. GFRP has lower tensile strength but higher flexural strength compared to CFRP [27]. Therefore, a hybrid laminated grid combining CFRP and GFRP is required when manufacturing the FRP grid to be used for structures that require bending strength.

### 4.2. Fracture Mode of the Composites

Figure 8 indicates the fracture mode of the composites after the pull-out test. The cutting process between the mortar and concrete was conducted to observe the destruction of the FRP grid and the surface fracture of the mortar. The red circle in the Figure 8 is the destroyed part of the FRP grid, and the blue line is the break line of the mortar. The crack in the mortar was not observed in the H.C and I.C specimens, and the CFRP grid was fractured in the load-bearing part. When the FRP grid is tensioned by a tensile load, the central axis of the FRP grid supports the dominant tensile stress, and the horizontal grid connected to it compresses the mortar by applying compressive stress to it. If the compressive strength of the mortar is higher than the compressive load applied by the horizontal grid, tensile failure of the FRP grid occurs before failure in the mortar specimen. The fractured part of the FRP grid embedded in the composites varies depending on the relationship between the tensile strength of the grid and the compressive strength of the mortar resisting the compressive stress of the horizontal grid. For the H.C and I.C specimens, the maximum tensile loads were 13.23 and 15.54 kN, respectively. The maximum loads applied to the H.C and I.C specimens were insufficient to cause compressive failure in the mortar, but the load could destroy the CFRP grid. The maximum tensile load of the P.C specimen was 20.02 kN, which was higher than that of H.C and I.C. Among them, P.C-3 showed the highest maximum tensile load of 20.77 kN. In the case of the P.C-3 specimen, it was observed that the mortar was destroyed before the destruction of the CFRP grid. Microcracks in the mortar were also observed in the P.C-1 and 2 specimens, but the damage to the CFRP grid was preferentially made such that the mortar was not destroyed.

Since the GFRP grid in the P.G specimen was destroyed at the maximum tensile load of 7.43 kN, the failure of the mortar did not occur. However, the failure of the GFRP grid occurred at the P1 grid point, not the load-bearing part. Since the FRP grids of all composite specimens were fabricated with the same manufacturing process of the composites and pull-out test method, it is assumed that the effect of FRP grid bending is equal. However, the GFRP grid of the P.G specimen has a relatively stronger bending resistance than the H.C and I.C specimen due to the high bending stiffness properties of glass fiber. Therefore, the CFRP grids of the H.C and I.C specimens were destroyed at the load-bearing part where the highest bending stress was applied, whereas in the case of the P.G specimens, failure occurred at the P1 grid point, the starting point of the grid.

### 4.3. Partial Tensile Strain of FRP Grid

Figure 9 shows the result of the partial tensile strain change of the FRP grid. The change in strain was measured when the load was increased at five grid points from P1 to P5, close to the load-bearing part. The measurements of strain values of the P1, P2 grid point of the P.C-1 specimen and the P1 grid point of P.G-1 were terminated before reaching the maximum load. For all other specimens, strain measurements were stopped before the maximum load was reached, as the strain value was above 4000 μm/m. When the partial tensile strain of the FRP grid to which the strain gauge was attached exceeded 4000 μm/m, the measurement result was analyzed, assuming that the measurement was terminated due to the damage of the gauge. In further research, a strain gauge with a wider measuring range should be used.

Among the three specimens for each type, the specimen having the maximum load value in the middle was designated as the representative specimen of the type. Representative specimens are H.C-1, I.C-2, P.C-1 and P.G-1 specimens. The partial tensile strains of the H.C-1, I.C-2, and P.C-1 specimens sequentially decreased at each grid point. The partial tensile strain of the P1 grid point showed a linear value compared to the partial tensile strain of other grid points. The partial strain of P2~P5 grid points showed a phenomenon by which the deformation occurred as the load increased. The partial tensile strain of the FRP grid increased with the distance from the load-bearing part, the load at which the deformation began. At a low-tensile-load section, the deformation mainly occurred at the upper part of the FRP grid. As the tensile load increased, the deformation of the FRP grid was transmitted to the lower part of the FRP grid. Among them, in the case of the P.C-1 specimen, the distribution of load between the P1 grid point and other grid points was uniform compared to H.C-1 and I.C-2 specimens. In the case of the P.C-1 specimen, the load was transmitted relatively uniformly, dispersing the stress concentration in the load-bearing part, and it was able to withstand high displacement and high load compared to other specimens.

In the P.G-2 specimen, the tensile load was hardly transmitted at the P3~P5 grid points before reaching the maximum tensile load. Compared to other specimens, the load-bearing rate of the P1 grid point was relatively high. As can be seen from the surface fracture figures, all the P.G specimens fractured at the P1 grid point. The stress concentration of the P1 grid point was also related to the relatively low tensile strength of GFRP grid. At a relatively low tensile load of 4 to 7 kN, the CFRP grid showed a slight change in strain at the P3~P5 grid points. In order to transmit the tensile load to the P3~P5 grid points, a tensile load of approximately 7–8 kN must be applied, but the low tensile strength of the GFRP grid failed in transmitting the tensile load.

### 4.4. Tensile Strain Distribution of FRP Grid

Figure 10 and Figure 11 indicate the distribution of the partial tensile strain of the grid points for each tensile load and the load-bearing rate of each grid point. The load-bearing rate of each grid was calculated according to the following Equations (1) and (2) [24].
(1)$$S=\frac{A_{i}}{\sum_{i=1}^{n}A_{i}}\times 100\%$$
(2)$$A_{i}=\frac{1}{2}\left(\varepsilon_{i-1}+\varepsilon_{i}\right)h_{i}$$

Here, *S* is the load-bearing rate of the grid point, *A*_i_ is the strain value distributed to the grid points, *ε_i_* is the partial tensile strain of each grid point, and *h_i_* is the height of the FRP grid between each grid point.

The load-bearing rate of each grid point was proportional to the height from the load-bearing part as it reached the maximum tensile load. In the case of the CFRP grid specimens, at a low load of 2 kN, the load-bearing rate of the P1 grid point exceeded 80%, and as the load increased, the load-bearing rate of the P1 grid point decreased. When a load of 12 kN was applied to the CFRP grid, the load-bearing rate of the P1 grid point of H.C-1, I.C-2 and P.C-1 was 56.23%, 56.8%, and 52.08%, respectively. When a load of 14 kN was applied to the CFRP grid, the load-bearing rate of the P1 grid point of the H.C specimen was 50.40% and that of the P.C specimen was 47.51%. When a load of 16 kN was applied to the CFRP grid, the load-bearing rate of the P1 grid point of the P.C specimen was 42.83%. If the tensile strength of the FRP grid is improved to withstand a higher maximum tensile load, the partial tensile strain of the P1 grid point is decreased and the tensile load can be evenly distributed throughout the FRP grid. For the P.G specimen, the load-bearing rate of the P1 grid point exceeded 80% under all measured tensile loads. The maximum tensile load of the P.G-2 specimen was 7.43 kN, but the measurement was terminated around the load reaching 5.8 kN, such that the load data of the P.G-2 specimen were analyzed up to 5 kN. When a load of 5 kN was applied to the GFRP grid, the load-bearing rate of the P1 grid point was 81.45%. However, under the load of 5 kN of the CFRP grid specimens, the load-bearing rate of P1 of H.C-1, I.C-2 and P.C-1 was 83.18%, 81.66%, and 74.79%, respectively, showing high values as with the GFRP grid. Due to the low tensile strength of the GFRP grid, failure occurred before the load borne by the P1 grid point was transferred to the other grids.

Figure 12 shows the distribution of partial tensile strain of the specimen under a low load of 2 kN, and a high load of 12 kN was investigated. In the case of the P.G specimen, the maximum tensile load was 7.43 kN; thus, it was excluded from the strain data with a load of 12 kN. The distribution of the four specimens at 2 kN and the strain distribution of the three specimens at 12 kN showed similar load distributions, respectively. The magnitude of the applied load has a dominant influence on the strain distribution of the FRP grid embedded in composites. In addition, as the magnitude of the load increased, the loads initially concentrated on the P1 grid point were distributed to the lower grid points.

### 4.5. Finite Element Analysis of CFRP/GFRP Grid

The geometry and boundary conditions of the model used for the finite element analysis (FEA) are shown in Figure 13a,b. ABAQUS commercial tools were used for finite element analysis. The finite element model uses two-dimensional and symmetric conditions to increase the efficiency of analysis. For the finite element model, grid parts and mortar parts were applied, and the concrete was modeled as boundary conditions. The element size is 3 mm × 3 mm, and the element type is C3D20R (20-node quadratic brick, reduced integration). Static general analysis was conducted and strain in the load direction was derived from grid points P1 to P5. For the analysis, two models, P.C and P.G types were set as the analysis case and analysis was performed.

As a result of the finite element analysis, the strain values of the two grid materials were similar to the experimental results. In the same mortar condition, the experimental and finite element analysis results of the P.C type showed that the stiffness was about 2.8 times higher than that of the P.G type. Figure 13c shows the tensile strain distribution of the grid when the tensile load is applied to the load-bearing part. It can be seen that the tensile stress is concentrated in the load-bearing part, and this is the reason that most of the fracture surfaces of the FRP grid in Figure 8 are concentrated on the load-bearing part. As for the distribution of tensile strain, in the P1 grid point, the strain is distributed in the horizontal grid, but from the P2~P5 grid point, the strain in the horizontal grid is insignificant. If the maximum tensile load is increased by improving the rigidity of the grid, strain distribution in the horizontal grid is likely to occur. Figure 14 shows the distribution of tensile strain in each grid. Comparing the experimental and analysis results of the tensile strain distribution showed similar results. Similar to the experimental results, in the analysis results, P1 grid point showed the highest load-bearing rate, and the rate decreased as the distance from the load-bearing part increased. The experimental result showed that the dispersion of strain in P1 was more concentrated than the analysis result, which is interpreted as the effect of the bending stress applied during the tensile test.

## 5. Conclusions

In the present study, three types of CFRP grid and one type of GFRP grid were fabricated using three fabrication methods. The tensile behavior of the FRP grid incorporated into the FRP-grid-reinforced concrete was investigated. The effect of the fabrication method for the FRP grid and type of sheets on the tensile strength of the FRP grid was analyzed. The results of load extension, fracture mode, partial tensile strain, and load-bearing rate of each grid point are summarized as follows:(1)The load extension results of the pull-out test, including the maximum tensile strain, stress, and load, were evaluated. The tensile behavior of the FRP grid was affected by the difference in fiber impregnation rate and inter-sheet adhesion, according to the processing process. The P.C specimen manufactured by the prepreg OVB method showed 15.68% and 8.14% higher displacements and 56.38% and 7.42% higher maximum loads compared to the H.C and I.C specimens, respectively.(2)The fracture shapes of the composites after the pull-out were analyzed. The fractured part of the FRP grid embedded in composites varied depending on the relationship between the tensile strength of the grid and the compressive strength of the mortar resisting the compressive stress of the horizontal grid.(3)The partial tensile strain of each grid point was investigated. The partial tensile strain of the FRP grid increased with the distance from the load-bearing part, the load at which the deformation began. At a low-tensile-load section, the deformation mainly occurred at the upper part of the FRP grid. As the tensile load increased, the deformation of the FRP grid was transmitted to the lower part of the FRP grid. The FRP grid embedded in composites did not undergo constant deformation but had different deformations in parts.(4)The load-bearing rate of each grid point was evaluated. The load-bearing rate of each grid point was proportional to the height from the load-bearing part when reaching the maximum tensile load. The magnitude of the applied load had a dominant influence on the strain distribution of the FRP grid embedded in composites.(5)Finite element analysis of FRP grid were conducted by using ABAQUS commercial tools. As a result of the finite element analysis, the strain values of the two grid materials were similar to the experimental results. Analysis results indicated that the tensile stress is concentrated in the load-bearing part. P1 grid point showed the highest load-bearing rate, and the rate decreased as the distance from the load-bearing part increased at FEA results.

This study considered the mechanical effects of the FRP grid fabrication method for the design guide of the FRP grid that can be used for concrete reinforcement. In the further research, it is necessary to review the drawing method for fabrication of the FRP grid, and additional consideration on the mix proportion of the FRP laminate. In addition, a finite element analysis on the mechanical performance according to the shape of the FRP grid will be performed.

## Figures and Tables

**Figure 1 materials-15-00357-f001:**
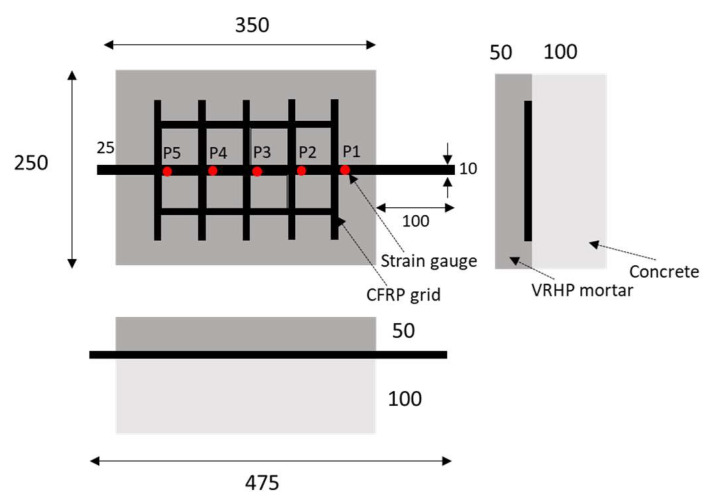
Schematics of FRP grid/concrete/VRHP mortar composites (Units: mm).

**Figure 2 materials-15-00357-f002:**
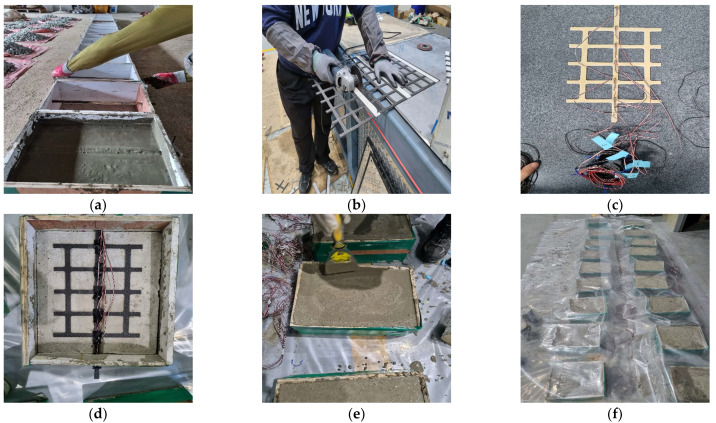
Specimen fabrication of FRP-grid-reinforced concrete with VRHP mortar. (**a**) Concrete casting. (**b**) FRP grid fabrication. (**c**) Installation of strain gauge. **(d**) FRP grid installation on. concrete. (**e**) VRHP mortar casting. (**f**) Curing FRP grid reinforced concrete.

**Figure 3 materials-15-00357-f003:**
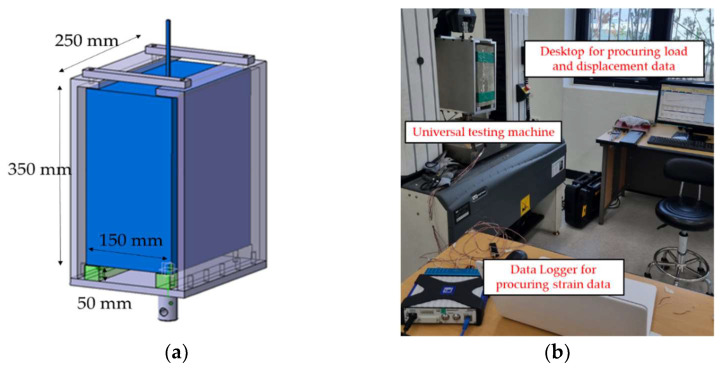
(**a**) Schematic of jig for holding the composites; (**b**) pull-out test setup.

**Figure 4 materials-15-00357-f004:**
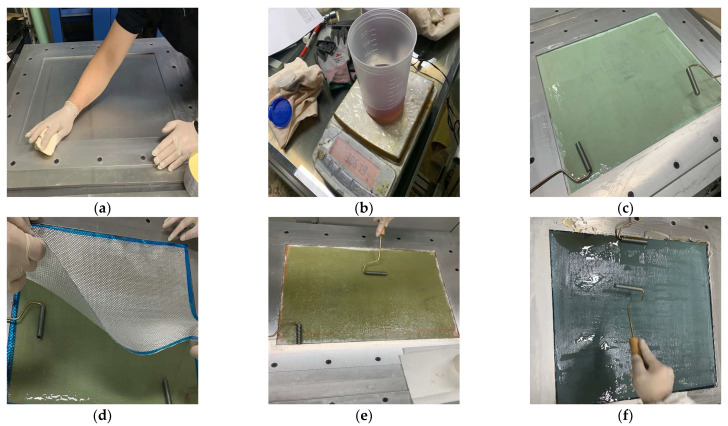
Specimen fabrication by hand lay-up method. (**a**) Surface release treatment. (**b**) Mixing resin. (**c**) Inner lamination of peel ply. (**d**) Fabric lamination. (**e**) Adhesive impregnation by hand rolling. (**f**) Lamination of peel ply and release film.

**Figure 5 materials-15-00357-f005:**
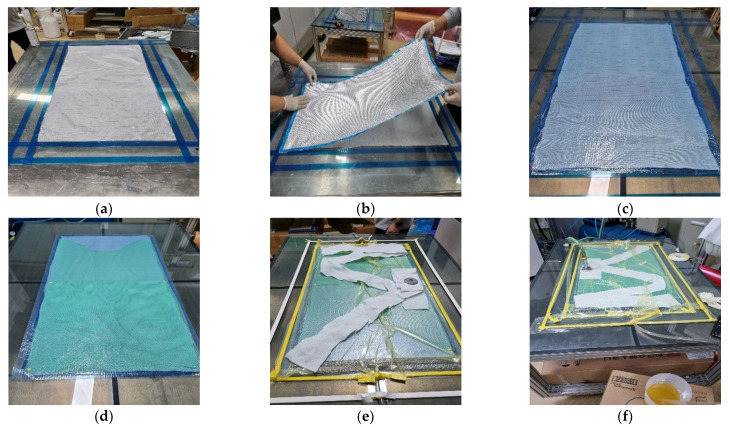
Specimen fabrication by resin infusion method. (**a**) Peel ply lamination. (**b**) Fabric lamination. (**c**) Release film lamination. (**d**) Mesh lamination. (**e**) Installation of omega flow and resin port. (**f**) Resin injection after vacuum bagging.

**Figure 6 materials-15-00357-f006:**
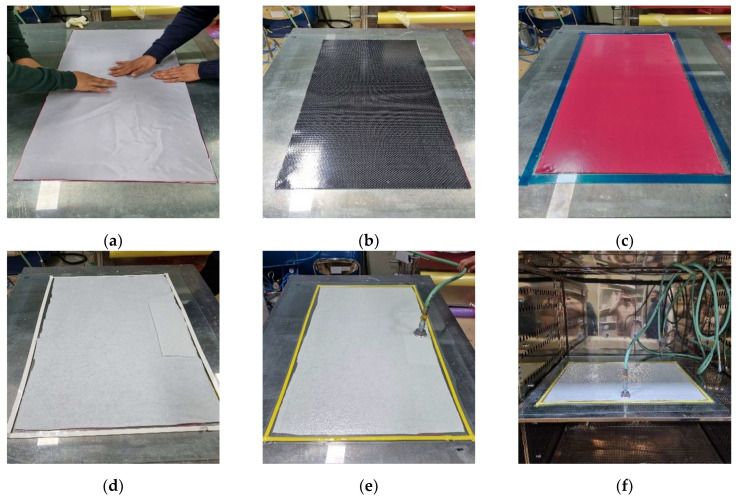
Specimen fabrication by prepreg OVB method. (**a**) Peel ply lamination. (**b**) Prepreg lamination. (**c**) Lamination of adhesive film and release film. (**d**) Lamination of breather fabric and vacuum film. (**e**) Compaction by vacuum bagging. (**f**) Oven curing during vacuum bagging.

**Figure 7 materials-15-00357-f007:**
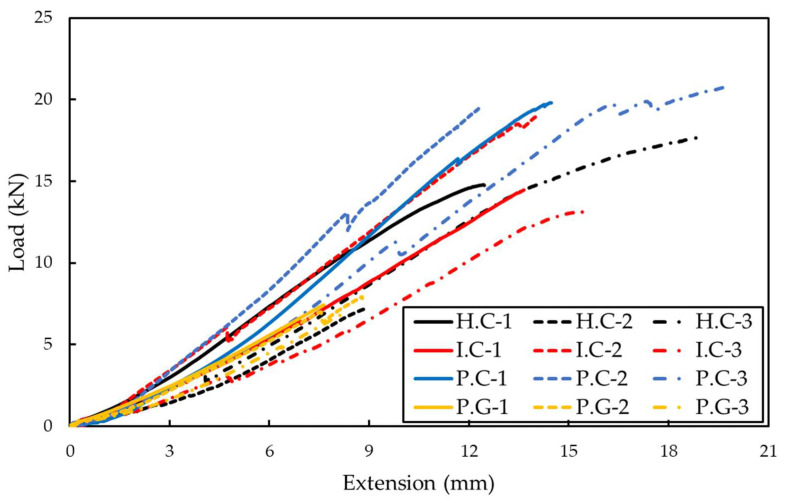
Load-Extension for CFRP/GFRP grid specimen.

**Figure 8 materials-15-00357-f008:**
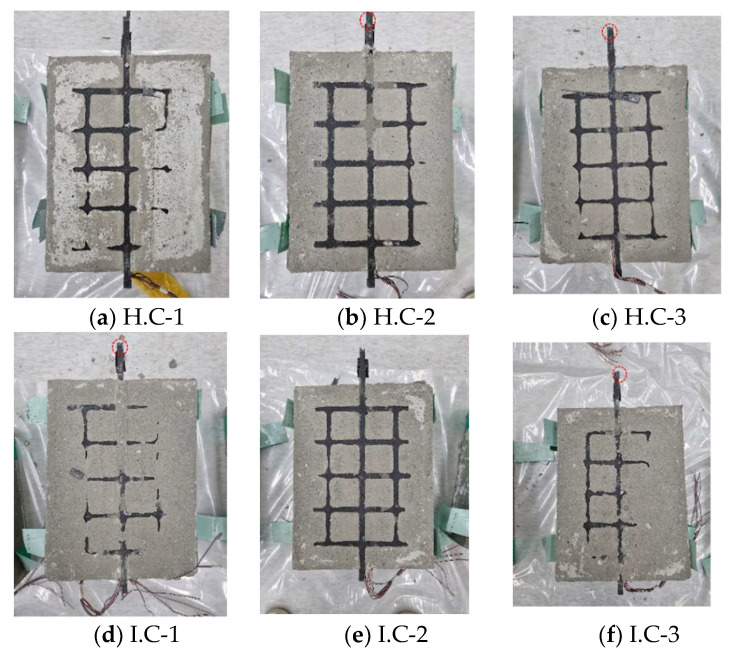
Surface fracture of specimens of CFRP/GFRP grid reinforced concrete.

**Figure 9 materials-15-00357-f009:**
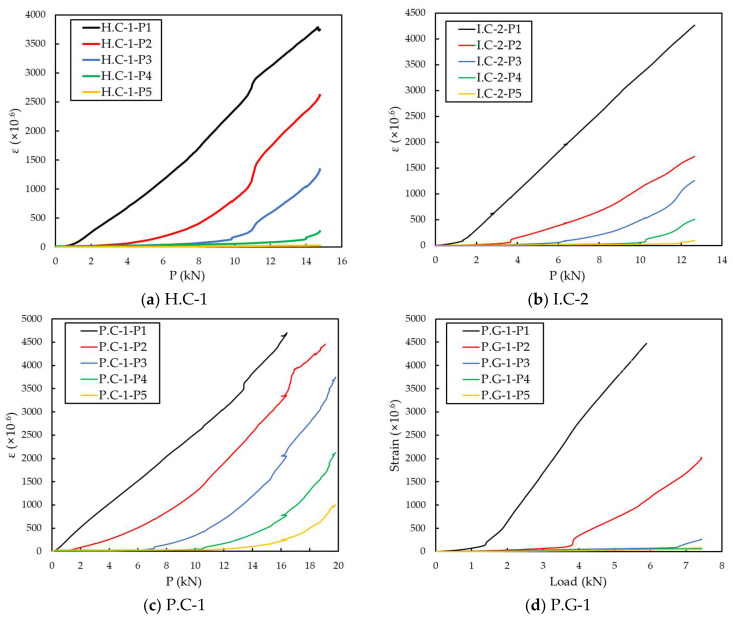
Partial tensile strain for CFRP/GFRP grid points.

**Figure 10 materials-15-00357-f010:**
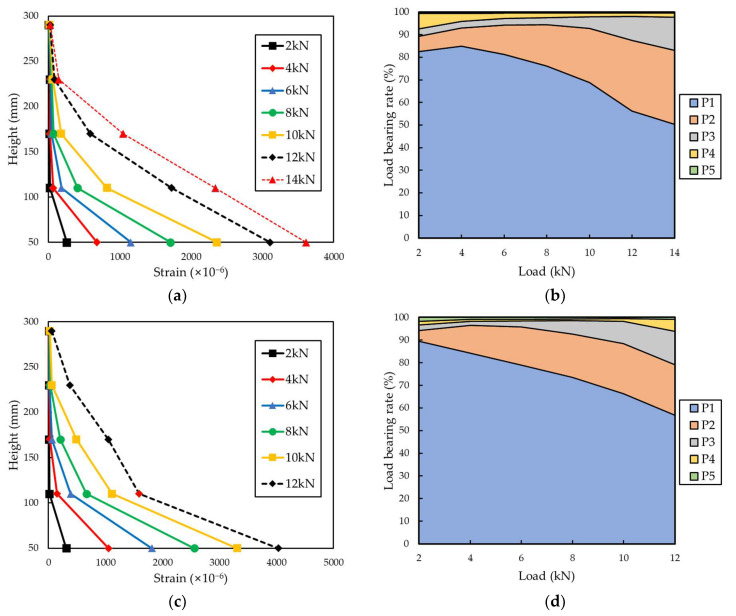
Tensile strain distribution and load-bearing rate of each point of H.C-1 and I.C-2. (**a**) Tensile strain distribution of H.C-1. (**b**) Load-bearing rate of each point of H.C-1. (**c**) Tensile strain distribution of I.C-2. (**d**) Load-bearing rate of each point of I.C-2.

**Figure 11 materials-15-00357-f011:**
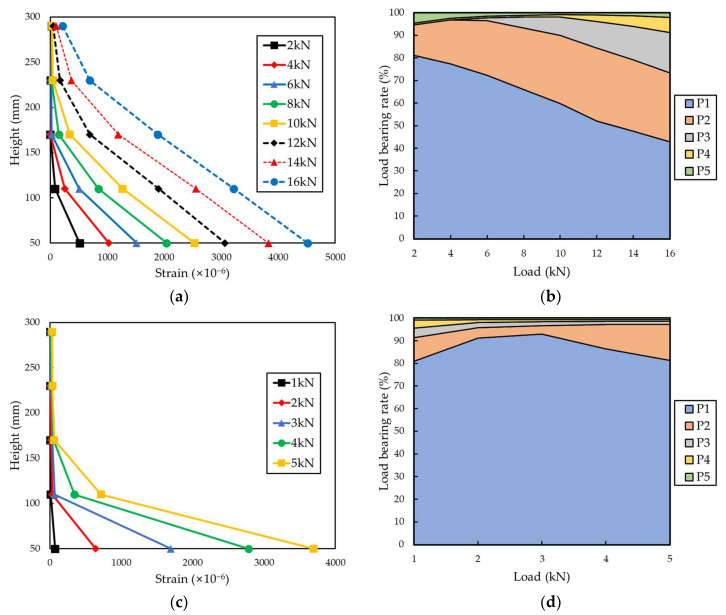
Tensile strain distribution and load-bearing rate of each point of P.C-1 and P.G-1. (**a**) Tensile strain distribution of P.C-1. (**b**) Load-bearing rate of each point of P.C-1. (**c**) Tensile strain distribution of P.G-1. (**d**) Load-bearing rate of each point of P.G-1.

**Figure 12 materials-15-00357-f012:**
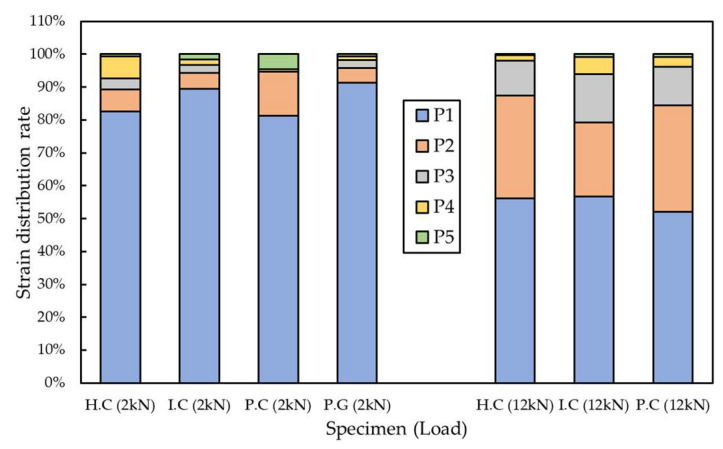
Load-bearing rate of grid points (2 kN, 12 kN of load).

**Figure 13 materials-15-00357-f013:**
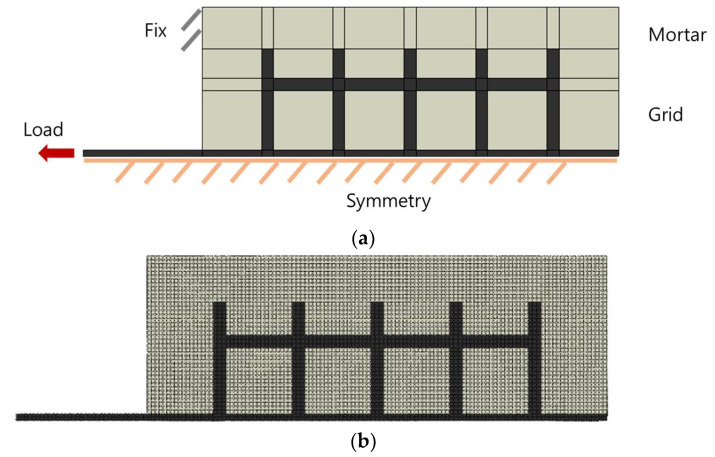
(**a**) Boundary condition, (**b**) finite element model, and (**c**) strain results of CFRP grid.

**Figure 14 materials-15-00357-f014:**
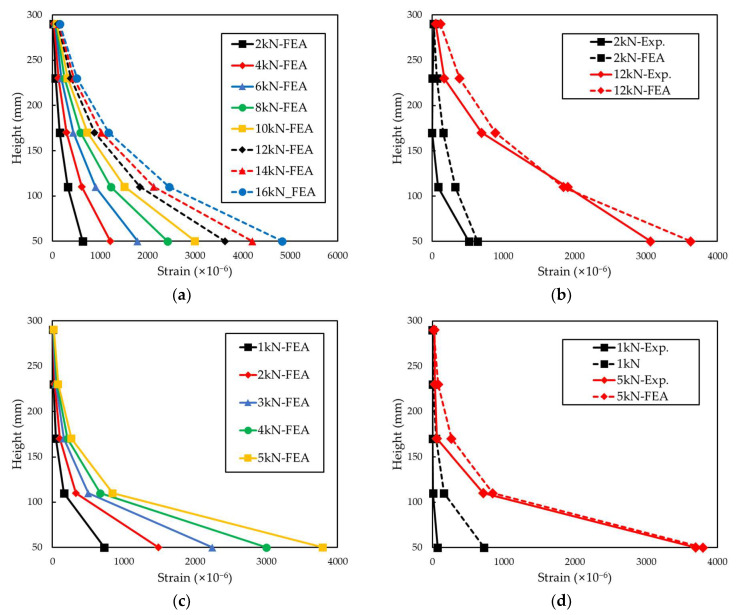
Comparison of tensile strain distribution of P.C and P.G specimen. (**a**) Tensile strain distribution of P.C specimen: FEA results. (**b**) Exp. vs. FEA results. (**c**) Tensile strain distribution of P.G specimen: FEA results. (**d**) Exp. vs. FEA results.

**Table 1 materials-15-00357-t001:** Carbon and glass fiber-reinforced plastics (CFRP/GFRP) laminate properties.

No.	Sheet	Ply	Nominal Weight (gsm)		Density (g/mm^3^ × 10^−3^)		Impregnation Rate of Fiber (wt. %)
			Fiber	Resin	Fiber	Resin	
1	Carbon fabric	6	420	344	1.78	1.20	55
2	Carbon fabric	6	420	180	1.78	1.20	70
3	Carbon prepreg	6	400	216	1.78	1.20	57.4
	Adhesive film	2	-	244	-	1.20	
4	Glass prepreg	10	323	174	2.54	1.20	59.2
	Adhesive film	2	-	244	-	1.20	

**Table 2 materials-15-00357-t002:** Type of CFRP/GFRP grid specimens.

Specimen Code	Sheet	Method	Thickness (mm)	Width (mm)
H.C	Carbon fabric	Hand lay-up	3.1	10
I.C	Carbon fabric	Resin infusion	2.6	
P.C	Carbon prepreg	OVB	2.8	
P.G	Glass prepreg	OVB	3.1	

**Table 3 materials-15-00357-t003:** Mix proportion of concrete (kg/m^3^).

W/B	Cement	Sand	Gravel	Water	Admixture
0.5	457	686	1086	229	23

**Table 4 materials-15-00357-t004:** Tensile properties of CFRP/GFRP grid specimen under pull-out test.

Specimen	Ultimate Tensile Strain (%)	Maximum Load (kN)	Ultimate Tensile Stress (MPa)
H.C	3.84	13.23	402.04
I.C	4.10	15.54	585.30
P.C	4.44	20.02	628.73
P.G	2.35	7.43	224.82

## Data Availability

No new data were created or analyzed in this study. Data sharing is not applicable to this article.

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
