# Peer review of "A Study on Tensile Behavior According to the Design Method for the CFRP/GFRP Grid for Reinforced Concrete"

_materials, 2022, doi:10.3390/ma15010357_

Round 1

Reviewer 1 Report

This paper is of great engineering significance and plays an important reference role in the design and manufacture methods of CFRP / GFRP grid. However, for it to be published, in my opinion, a few things need to be improved.

Author Response

Response to Reviewer 1 Comments

We have revised our manuscript in accordance with the reviewers’ comments to improve the manuscript. We are sincerely grateful to the reviewers for his elaborated comments. We respond to each review comment or suggestion made by the reviewers in detail below.

Point 1: The form of acronyms needs to be corrected. The full name should be indicated when it appears for the first time, and there is no need to explain or use the full name when they appear later.

Response 1: The format of all acronyms has been corrected according to the reviewer's comments.

Point 2: The  introduction  is  not  comprehensive  enough  and  lacks  a  description  of  the  existing manufacturing methods of FPR grid.

Response 2: To address the reviewer’s comment, the following sentences were added to the revised manuscript:

“The manufacturing method of FRP is selected considering the purpose and cost of FRP composites. In FRP molding, the final product is manufactured by direct molding the fabric, or through secondary molding after fabricating an intermediate material such as prepreg. Representative direct molding methods include hand lay-up and resin infusion methods. Hand lay-up method is used for simple and low-volume products, and is applied for boat shells, swimming pools, and bathtubs. Resin infusion method is mainly used to make large structures such as wind blades. In the prepreg manufacturing method, a final product is manufactured by compaction of pre-prepared prepregs mainly through the oven vacuum bagging method [11–16]. The detailed manufacturing method of the above three methods will be dealt with in Chapter 3”

(57th thru 66th lines from the manuscript of the revision)

Point 3: Please  unify  the  expression  of  hand  lay-up  method  in  this  paper.  “hand  lay-up” or “hand-layup”?

Response 3: The expression of hand lay-up method was unified as “hand lay-up” in the revision.

Point 4: Please check and correct the unit of “Ultimate tensile strain” in Table 5

Response 4: The unit of “ultimate tensile strain” is expressed as % in Table 5.

Point 5: In Section 3.3, Please explain the reasons for only selecting these four specimens for analysis. There are two different types of specimen numbers in Figure 10 (c) and (d). The names of coordinate in Figure 10 (d) are inconsistent with those in the previous three figures. Please check and correct them.

Response 5: The specimen numbers in Figure 10 (c) and (d) have been corrected and to reflect the reviewer’s comment, the following was added to the revised manuscript:

“Among the three specimens for each type, the specimen having the maximum load value in the middle was designated as the representative specimen of the type. Representative specimens are H.C-1, I.C-2, P.C-1 and P.G-1 specimens.”

(391st thru 393rd lines from the manuscript of the revision)

Point 6: The finite element analysis part was mentioned in the abstract and introduction. It seems that it did not clearly state which part is finite element analysis, and there is a lack of expression of the model and the effective process of verifying the model.

Response 6: Finite element analysis has been completed. The results of finite element analysis have been added to Chapter 4.5 of the revision.

Point 7: Please refer in the conclusion to the quantitative assessment of the research conducted.

Response 7: The Quantitative assessment was added to the conclusion according to the reviewer's comments.

Point 8: The number of the subtitle of section 4 is incorrect. Please check and correct it.

Response 8: The number of the subtitle of section 4 has been corrected.

Point 9: Page5, Line 155, The word "smoothes" should be replaced by “smooths”.

Response 9: The word “smoothes” has to be replaced by “smooths” in the revision.

Reviewer 2 Report

The article covers the topic of a study on the tensile behavior according to the design method of the CFRP/GFRP grid for reinforced concrete 
This is very interesting and informative paper .
In general the manuscript has acceptable cohesion. 
In my opinion the following modification must be included before further potential processing:

1. It is recommended to show major results in the abstract part (what was achieved in general?).
2. I suggest to add separated point - Research significance - Please describe here the main essence of the research. 
What was the inspiration for such an analysis? Why presented studies are so important?
I suggest to use text in line 80-87.
3. I suggest that in introduction part, some valuable information about general types of FRP materials should be added.
In my opinion it is worth to describe shortly (in 1 - 2 sentences) more major materials in this technique (CFRP, BFRP, GFRP and AFRP).
4. Table 2 - please explain specimen codes.
5. The morphology between composite and concrete (mortar) is very improtant aspect.
Please determine some major aspects influences on the quality of this joint.
6. Please add a scale in figures 3 and 4a.
7. Please add cement content and determine size of aggregates (sand and coarse aggregate).
8. In conclusion part it is recommended to indicate potential application of research results in engineering. 

Author Response

Response to Reviewer 2 Comments

We have revised our manuscript in accordance with the reviewers’ comments to improve the manuscript. We are sincerely grateful to the reviewers for his elaborated comments. We respond to each review comment or suggestion made by the reviewers in detail below.

Point 1: It is recommended to show major results in the abstract part (what was achieved in general?).

Response 1: To address the reviewer’s comment, the following sentences were added to the revised abstract:

“The CFRP grid manufactured by the prepreg OVB method showed the best tensile behavior compared to the CFRP grid manufactured by the hand lay-up and resin infusion methods. The load-bearing of each grid point was proportional to the height from the load-bearing part when reaching the maximum tensile load.”

(16th thru 20th lines from the manuscript of the revision)

Point 2: I suggest to add separated point - Research significance - Please describe here the main essence of the research. What was the inspiration for such an analysis? Why presented studies are so important? I suggest to use text in line 80-87.

Response 2: To reflect the reviewer’s comment, the following sentences were added to the revised manuscript:

“The main essences of the present study are as follows : (1) Present a design guideline of CFRP and GFRP grids, (2) Investigation of tensile behavior of FRP grid incorporated in FRP grid/concrete/VRHP mortar composites, (3) A Study on tensile strain distribution of FRP grids and (4) Comparative analysis with experimental results through finite element analysis.”

(97th thru 102nd lines from the manuscript of the revision)

Point 3: I suggest that in introduction part, some valuable information about general types of FRP materials should be added. In my opinion it is worth to describe shortly (in 1 - 2 sentences) more major materials in this technique (CFRP, BFRP, GFRP and AFRP).

Response 3: To address the reviewer’s comment, the following sentences were added to the revised manuscript:

“FRP composites include glass fiber reinforced plastics (GFRP), basalt fiber reinforced plastics (BFRP), aramid fiber reinforced plastics (AFRP) and carbon fiber reinforced plastics (CFRP). Glass fiber made from thin glass has good cost effectiveness, good impact resistance, and good fire resistance, but its weight reduction efficiency is low [17]. Basalt fiber made by melting basalt at high temperature is an eco-friendly fiber that does not generate carcinogens, and has good mechanical properties compared to glass fiber, but its weight reduction efficiency is low. Aramid fiber has a flame retardant, good elongation, and impact resistance, but is expensive and have poor resin adhesion rate [18].”

(43rd thru 51st lines from the manuscript of the revision)

Point 4: Table 2 - please explain specimen codes.

Response 4: To reflect the reviewer’s comment, the following sentences were added to the revised manuscript:

“Specimen codes are defined as follows: The first letter ‘H’ stands for hand lay-up method, ‘I’ stands for resin infusion method, and ‘P’ stands for prepreg OVB method. The second letter ‘C’ stands for carbon fabric and ‘G’ stands for glass fabric.”

(114th thru 117th lines from the manuscript of the revision)

Point 5: The morphology between composite and concrete (mortar) is very improtant aspect. Please determine some major aspects influences on the quality of this joint.

Response 5: To address the reviewer’s comment, the additianal information of the morphology beteween concrete and mortar were added to the revised manuscript:

“After curing of the concrete specimen was completed, the surface was smoothed using sandpaper. The FRP grid was attached on a concrete specimen by a surface attachment method without using a rivet anchor because the overall size of the composite was not large. [24]. Strain gauges were installed at P1~P5 grid point of FRP grid. The strain gauge was coated with silicone bond and rubber tape to prevent external impact.”

(131st thru 135th lines from the manuscript of the revision)

Point 6: Please add a scale in figures 3 and 4a.

Response 6: The scale in figures has been added.

Point 7: Please add cement content and determine size of aggregates (sand and coarse aggregate).

Response 7: The content of cement is specified in Table 3 and the following sentence was added to the revised manuscript:

“River Sand (0.06 ~ 2 mm) and gravel (9.5 ~ 37.5 mm) were used as fine and coarse aggregates, respectively.”

(129st thru 130th lines from the manuscript of the revision)

Point 8: In conclusion part it is recommended to indicate potential application of research results in engineering

Response 8: To reflect the reviewer’s comment, the following sentences were added to the revised manuscript:

“This study considered the mechanical effects of the FRP grid fabrication method for the design guide of the FRP grid that can be used for concrete reinforcement. In the further research, it is necessary to review the drawing method for fabrication of the FRP grid, and additional consideration on the mix proportion of the FRP laminate. In addition, a finite element analysis on the mechanical performance according to the shape of the FRP grid will be performed.”

(526th thru 531st lines from the manuscript of the revision)

Round 2

Reviewer 2 Report

All remarks have been considered by authors.

I suggest that paper could be published in 

current form.